# Comparison of Modern In Vitro Permeability Methods with the Aim of Investigation Nasal Dosage Forms

**DOI:** 10.3390/pharmaceutics13060846

**Published:** 2021-06-08

**Authors:** Csilla Bartos, Piroska Szabó-Révész, Tamás Horváth, Patrícia Varga, Rita Ambrus

**Affiliations:** Faculty of Pharmacy, Institute of Pharmaceutical Technology and Regulatory Affairs, University of Szeged, 6720 Szeged, Hungary; revesz@pharm.u-szeged.hu (P.S.-R.); horvath.tamas@pharm.u-szeged.hu (T.H.); varga.patricia@szte.hu (P.V.); ambrus.rita@szte.hu (R.A.)

**Keywords:** nasal administration, Side-Bi-Side diffusion cell, Franz diffusion cell, in vitro permeability

## Abstract

Nowadays, the intranasal route has become a reliable alternative route for drug administration to the systemic circulation or central nervous system. However, there are no official in vitro diffusion and dissolution tests especially for the investigation of nasal formulations. Our main goal was to study and compare a well-known and a lesser-known in vitro permeability investigation method, in order to ascertain which was suitable for the determination of drug permeability through the nasal mucosa from different formulations. The vertical diffusion cell (Franz cell) was compared with the horizontal diffusion model (Side-Bi-Side). Raw and nanonized meloxicam containing nasal dosage forms (spray, gel and powder) were tested and compared. It was found that the Side-Bi-Side cell was suitable for the investigation of spray and powder forms. In contrast, the gel was not measurable on the Side-Bi-Side cell; due to its high viscosity, a uniform distribution of the active substance could not be ensured in the donor phase. The Franz cell, designed for the analysis of semi-solid formulations, was desirable for the investigation of nasal gels. It can be concluded that the application of a horizontal cell is recommended for liquid and solid nasal preparations, while the vertical one should be used for semi-solid formulations.

## 1. Introduction

Intranasal administration is an effective way to deliver drugs into the systemic circulation or brain tissues as an alternative to the oral and parenteral routes for some therapeutic agents [1,2]. Powders, sprays and gels are the official pharmacopoeial nasal dosage forms [3]. Administration through the nasal route has several advantages, such as the rapid onset of action, the circumvention of the first-pass elimination by the liver and the gastrointestinal (GI) tract, the non-invasiveness, but nonetheless, simple daily administration [4]. The intranasal application of nonsteroidal anti-inflammatory drugs (NSAIDs) may be an alternative route for acute pain therapy or to enhance analgesia. In our previous works, meloxicam- (MX; a poorly water soluble NSAID) containing formulations were developed, intended for intranasal administration. In all cases, the bioavailability of MX was increased by decreasing its particle size [5,6,7]. In the design of an intranasal powder or suspension as a drug delivery system, it is important to consider the requirement for the particle size of the product (from 5 to 40 µm) [8].

Beyond the physicochemical properties of drugs (such as solubility, molecular weight, lipophilicity, pKa, etc.), nasal transmucosal absorption is affected by the properties of the nasal cavity. Because of its limited volume, only small doses may be applied (1–200 µL) [9]. This can cause challenges for the development, and also narrows down the number, of the useable drug candidates. Moreover, the nasal respiratory epithelium consists of columnar cells covered with cilia, which move in a coordinated way to propel mucus from the epithelial surface towards the pharynx. This coordinated move is called mucociliary clearance, which has one of the strongest effects on nasal drug administration. Depending on the beat frequency of the cilia, the administrated dose can stay only for 10–15 min on the nasal mucosa until it is renewed [10,11].

Formulation factors, such as dosage form, pH, viscosity, mucoadhesivity, also influence [12,13,14] the drug diffusion through the nasal mucosa. To overcome the short residence time of the drugs in the nasal cavity, as a result of mucociliary clearance, different mucoadhesive excipients (e.g., chitosans, lectins, thiomers, poloxamer or sodium hyaluronate) may be used during the formulation, in order to prolong the contact time with the nasal mucosa, thereby enhancing the delivery of the drugs [15,16,17]. Furthermore, the usage of mucoadhesive additives allows the delivery of nanosized drugs intranasally to overcome the limitations of the particle size requirements for nasal formulations. Based on our previous experience, it can be concluded that the bioavailability of an active pharmaceutical ingredient is increased by decreasing its particle size [18,19].

For the investigation of nasal formulations, screening methods are used which are able to give information about nasal preparations. A general penetration investigation protocol of a pharmaceutical nasal composition includes in vitro, in vitro cell line, ex vivo and in vivo investigations. In vitro studies are carried out using artificial membrane and the development of in vitro models is of great importance in the simplification and quickening of animal studies [20]. Several solutions and methods were carried out to imitate the nasal environment and help to predict the behavior of the tested forms. Donor−acceptor diffusion experiments represent the typical set up for in vitro permeation studies. Usually the diffusion chambers are well defined by their volume and geometry. The typical volumes of the donor or acceptor compartments are in the range of 1 and 12 mL. In conventional donor−acceptor methods, the stirring of the buffer media is achieved by means of magnetic bars [21]. The μFLUX diffusion Cell (Pion Inc., Billerica, MA, USA) is a horizontal diffusion system in which the compartments are divided by a synthetic membrane and may be suitable for the investigation of nasal dosage forms. By the installation of a UV detector, a real-time concentration measurement is allowed on both sides. The Navicyte Vertical Ussing Diffusion Chamber System (Harvard Apparatus, Holliston, MA, USA) is suitable for experiments on biological materials (cells or tissues) and there are literature data about its application for the investigation of intranasal formulations. The Navicyte Horizontal Diffusion Chamber System (Harvard Apparatus, Holliston, MA, USA) is designed for transport studies using cells and tissues which are exposed to an air interface in their normal in vivo environment such as nasal, pulmonary, corneal or dermal cells [22,23]. The Franz diffusion cell system is the official pharmacopoeial method, primarily for the investigation of the diffusion of transdermal formulations, but it is one of the most commonly used method for the investigation of intranasal dosage forms in the literature [24,25]. The horizontal Side-Bi-Side system has a small volume of donor and acceptor compartments allowing the measurement of small amounts of samples, as nasal dosage forms. Magnetic stirring of the donor phase imitates the movement of cilia in the nasal cavity [26].

Our aim was to compare two in vitro investigation models with different orientations of the phases; as a vertical system the Franz diffusion cell was selected and as a horizontal one the Side-Bi-Side system, from the aspect of their applicability for the investigation of different intranasal dosage forms: sprays, gels and powders. The nasal formulations contained MX as an active pharmaceutical ingredient (API). In order to increase the bioavailability of MX, it was nanonized by co-grinding in a planetary ball mill and for comparison, a physical mixture of MX and additive was used. The permeability results of different intranasal formulations—obtained on the vertical and horizontal cell systems—were compared and analyzed.

## 2. Materials and Methods

### 2.1. Materials

MX (4-hydroxy-2-methyl-N-(5-methyl-2-thiazolyl)-2H-benzothiazine-3-carboxamide-1,1-dioxide) was obtained from Egis Plc. (Budapest, Hungary) as a poorly soluble NSAID. The grinding additives, polyvinylpyrrolidone PVP-C30 (PVP) were purchased from BASF (Ludwigshafen, Germany). Sodium hyaluronate (HA) (Mw = 1400 kDa) was obtained as a gift from Gedeon Richter Plc. (Budapest, Hungary). Binary mixture of as-received MX powder and the carrier (PVP) was mixed.

### 2.2. Methods

#### 2.2.1. Preparation of the Co-Ground Product

Nanonized MX (nanoMX) was produced by planetary mono mill (400 rpm, 2 h, MX:PVP ratio = 1:1) (Retsch PM100 MA, Retsch GmbH, Haan, Germany). The details of the top-down method are available in our previous study [5]. The physical mixture of MX and PVP (MX/PVP mix powder) was produced as the control sample by a Turbula mixer (Turbula System Schatz; Willy A. Bachofen AG Maschinenfabrik, Basel, Switzerland) used at 60 rpm for 10 min in the same mass ratio (1:1) as the co-ground product.

#### 2.2.2. Characterization of Raw MX and Milled nanoMX

##### Size Distribution by Laser Diffraction

The particle-size distribution of the spray-dried samples was measured by laser scattering (Malvern Mastersizer Sirocco 2000, Malvern Instruments Ltd., Worcestershire, UK). The measurements were carried out at 3 bar pressure and 75% frequency, air was used as a dispersion medium. Approximately 1 g of product was tested in one measurement, and each measurement was performed 3 times. D0.1, D0.5 and D0.9 values were determined as the diameter of the particles below which 10, 50 and 90 volume percentage of the particles exist.

##### Scanning Electron Microscopy (SEM)

The shape and surface morphology of the spray-dried particles was visualized by SEM (Hitachi S4700, Hitachi Scientific Ltd., Tokyo, Japan). Under an argon atmosphere, the samples were sputter-coated with gold-palladium in a high-vacuum evaporator with a sputter coater and they were examined at 10 kV and 10 μA. The air pressure was 1.3–13 MPa.

##### X-ray Powder Diffraction (XRPD)

XRPD was performed to investigate the physical state of MX in the samples with a Bruker D8 Advance diffractometer (Bruker AXS GmbH, Karlsruhe, Germany) with Cu K λI radiation (λ = 1.5406 Å). The samples were scanned at 40 kV and 40 mA with an angular range of 3° to 40° 2θ. Si was used to calibrate the instrument. DIFFRACTPLUS EVA software was used to perform the manipulations: Kα2-stripping, background removal and smoothing.

#### 2.2.3. Preparation of Intranasal Forms

Intranasal formulations, sprays, gels and powders as dosage forms were developed, containing physical mixture (MX/PVP mix powder) or co-grinded powder mixture (nanoMX) (Table 1), which were in suspended form in liquid and semi-solid formulations. The spray formula contained 1 mg/mL MX and 1 mg/mL HA, while the HA concentration was adjusted to 5 mg/mL in the gel forms. HA was applied as a viscosity enhancer and mucoadhesive agent in order to prolong the residence time on the nasal mucosa [7,27]. Intranasal powders did not require any preparation.

#### 2.2.4. Comparison of Hanson’s Vertical Diffusion Cell and Side-Bi-Side Horizontal Diffusion Cell

During the permeability studies the dispersion media was phosphate buffer (PB) of pH = 5.60 for each formulation which was a mixture of stock solutions A and B. The 100 mL PB was made from 94.4 mL stock solution A (containing 9.08 mg/L KH2PO4) and 5.6 mL stock solution B (containing 11.61 mg/L K2HPO4 concentration). HA-containing samples were allowed to swell for 24 h in the media and these viscous matrixes served as vehicles for the distribution of nanoMX and MX/PVP mix powder. The acceptor media of the in vitro measurements was PB with pH = 7.40. It was made from NaCl (8.00 g/L), KCl (0.20 g/L), Na_2_HPO_4_·1H_2_O (1.44 g/L) and KH_2_PO_4_ (0.12 g/L), diluted up to 1000 mL with distilled water.

##### In Vitro Permeability on Hanson’s Vertical Diffusion Cell (Franz Cell) System

Franz cell (Hanson Microette Topical and Transdermal Diffusion Cell System) (Hanson Research, Chatsworth, Los Angeles, CA, USA) is a vertical diffusion cell, recommended for transdermal forms, primarily for gels and ointments. In Figure 1, the picture of Franz cell can be seen. The cell contains a donor and an acceptor phase, separated with a synthetic membrane (PALL Metricel membrane with 0.45 or 0.1 µm pores) (diffusion area: 1.8 cm^2^). The donor phase above is only 300 µL, while the acceptor phase at the bottom has a remarkably larger volume (7.0 mL) (Table 2). Due to the structure of the cell, only the acceptor phase can be stirred to provide the uniform distribution of the active substance. Several versions have been developed with different donor and acceptor phases, depending on the purpose of the application: special nail tester, ion-perphoretic tester and fiber optic equipped real-time active substance detector versions are available on the market.

During the in vitro permeability studies on a vertical Franz diffusion cell system, the cumulative amount of MX that diffused through a synthetic membrane from a nasal dosage form was measured against time. Powder, gel and spray forms were investigated. Only a few data are available in the literature about the investigation of powder forms on Franz cell, but in none of the studies were powders inserted directly into the donor phase.

The temperature of the phases was set to 37 °C, as the human body temperature. A dose of 300 µg of gel and spray form was delivered into the donor chamber by Hamilton pipette. Aliquots (0.8 mL) were taken from the acceptor phase by an autosampler (Hanson Microette Autosampling System) (Hanson Research, Chatsworth, CA, USA) at 5, 10, 15 and 60 min of the measurement and were replaced with fresh receiving medium. The diffused drug amount was determined spectrophotometrically (Unicam UV/VIS) at 364 nm. Six parallel measurements were carried out with each sample. The rotation of the stir-bar was set to 100 rpm.

##### In Vitro Permeability on Side-Bi-Side Horizontal Diffusion Cell System

The donor and the acceptor phase of the Side-Bi-Side diffusion cell (Crown Glass, New York, NY, USA) were connected in a horizontal direction (Figure 2). The two horizontal chambers were divided by an impregnated (with isopropyl myristate) synthetic membrane (PALL Metricel membrane with 0.45 or 0.1 µm pores), similar to the Franz cell. The volumes of the donor and the acceptor phase were the same (3.0 mL) with a 0.69 cm^2^ diffusion area (Table 2). Continuous stirring can be guaranteed in each chamber by a magnetic stirrer during the measurement time. In addition, several type of chambers have been developed to help the formulation: for example, the Valia−Chien Cell phase can imitate the surface of the cornea. The Side-Bi-Side cell is also suitable for real-time impedance measurement in both chambers.

Comparing in vitro permeability studies were carried out on a modified horizontal Side-Bi-Side cell model, similarly to the previous investigations on Franz cell. The calibration of the measurement (the delivery method and sample taking) were investigated and published in our previous article [28].

The temperature of the phases was 37 °C (Thermo Haake C10-P5, Merck KGaA, Darmstadt, Germany), as in the Franz cell measurement. Aliquots (2.0 mL) were taken from the acceptor phase by a pipette and were replaced with fresh receiving medium at 5, 10, 15 and 60 min of the measurement. The diffused amount of drug was determined spectrophotometrically. Each sample was measured three times. Spray and gel form were delivered into the donor chamber by Hamilton pipette. The solid powder form was washed into the donor phase with the buffer. The two chambers were divided by the hydrophilic mixed cellulose ester synthetic membrane (GN-6 Metricel MCE Membrane Filter, Merck KGaA, Darmstadt, Germany) impregnated with isopropyl myristate. The rotation of the stir-bar was set to 100 rpm.

##### Determination of the Flux and the Permeability Coefficient

The API flux (J) was calculated from the quantity of MX, which permeated through the membrane, divided by the insert membrane surface and the time duration [µg/cm^2^/h]. The permeability coefficient (K_p_ [cm/h]) was determined from J and the drug concentration in the donor phase (C_d_ [µg/cm^3^]) (Equation (1)):
(1)$$K_{p}=\frac{J}{C_{d}}$$

## 3. Results and Discussion

### 3.1. Characterization of Raw MX and Milled NanoMX

The raw MX was crystalline with an 85.39 µm average particle size in the MX/PVP mix powder, while the average particle size of the MX and PVP co-ground(nanoMX) was 140.4 (±69.2) nm. The XRPD measurements proved that due to the co-grinding process, the crystallinity of the MX was decreased, the typical characteristic peaks of MX disappeared. The SEM pictures showed large, hexagonal, smooth-surfaced crystals of raw MX (Figure 3a). Concerning the nanoMX powder, grinding in the presence of PVP was performed to obtain rounder, smoother surfaced amorphous nanoparticles (Figure 3b).

### 3.2. Comparison of the In Vitro Diffusion Methods

The most striking difference between the two investigated methods is the relative position of the chambers. In the Franz cell, the investigated formula penetrates in a vertical direction that is helped by gravitation. This diffusion direction is acceptable in semi-solid forms but not in the case of intranasal conditions.

In contrast with Side-Bi-Side system, the donor chamber of the Franz cell cannot be stirred, which is a very advantageous feature of the horizontal cell, when uniform drug distribution is required. Furthermore, stirring imitates the movement of the cilia of the nasal cavity.

Another important difference is the administration method of the dosage forms to the donor chamber. Administration can affect the quantity of the diffused material and the reproducibility of the diffusion measurement. In the Franz cell investigations, the formula is laid directly on the top of the membrane which helps the penetration to the acceptor phase. In contrast, in the case of Side-Bi-Side cell, the formulation is administered to the donor chamber, not to the membrane.

#### 3.2.1. Investigation of Nasal Spray Forms

The results of the permeability measurements of the spray forms with the same content (nanoMX spray and MX/PVP mix spray) were compared in the Franz diffusion cell system and Side-Bi-Side system (Figure 4). It can be seen that the highest diffused amount of MX (≈73 µg/cm^2^) and the quickest diffusion (due to the rapid dissolution of the drug) were observed in case of the nanoMX spray, measured on the horizontal diffusion cell. Initially, a rapid rising in the concentration was noticed (in first 5 min), followed by a less steeply increasing drug amount in the acceptor phase. The diffusion of the nanoMX spray was approximately at least two times higher than the others in 60 min. Beyond the better dissolution of amorphized nanoparticles [29,30], this observation can be explained by the uniform distribution of MX particles in the continuously stirred liquid medium, containing 1 mg/mL HA, as a mucoadhesive agent. The pH of the formulation (pH 5.6) and the low concentration of HA resulted in a sol state of samples. It should be excluded that the nanoMX particles (D0.5 = 140 nm) passed through the membrane without dissolving, because the membrane pore size (100 nm) was smaller than the particle size and the membrane was impregnated with isopropyl myristate [7]. The measurement of the spray forms was difficult on the Franz diffusion cell. Because of the low viscosity of the samples, keeping them on the horizontal surface of the membrane was problematic. Since the stirring was not resolved—as mentioned above—the measurement of the suspensions and sedimenting materials was limited. The results showed a steady increase in concentration and a lower permeated amount of MX compared with the results measured on the Side-Bi-Side system. This phenomenon could be explained partly with the fact that the Franz cell was developed for the investigation of semi-solid forms. On the other hand, the donor phase of Side-Bi-Side cell is suitable for inserting liquid formulations and due to the stirring with a magnetic stirrer, it could be useful for testing liquids and suspensions.

#### 3.2.2. Investigation of Gel Forms

The permeability of gels with the same contents (nanoMX gel and MX/PVP mix gel) were also compared on the Franz diffusion cell system and Side-Bi-Side system. Studying the samples, it can be seen that a much higher permeated drug concentration was determined in the case of the nanoMX gel applied with the Franz diffusion system (215 µg/cm^2^) (Figure 5). An almost ten-fold drug amount was diffused to the acceptor phase compared to the results of the measurements on the Side-Bi-Side cell, which could be explained, on the one hand by the rapid dissolution of nanosized and amorphized MX particles [31], which provided a faster diffusion and a higher drug concentration in the acceptor phase. On the other hand, the design of the Franz cell’s donor phase was developed for the investigation of semi-solid formulations [32,33]. The permeated drug amount of nanoMX gel on the Side-Bi-Side system was similar to the results of the physical mixture investigated on the Franz cell, and the lowest drug amount was detected in the case of MX/PVP mix gel on the Side-Bi-Side diffusion cell. This observation can be explained by the inadequate stirring in the donor compartment, furthermore, the nanoMX has a particle size similar to those of the polymeric molecules such as HA, PVP which can result in a well-structured complex, and better interactions among the components retaining MX from dissolution and preventing the drug from reaching the membrane surface [34,35]. It can be concluded that Franz diffusion system should be proposed for the investigation of intranasal gels.

#### 3.2.3. Investigation of Powder Forms

Our experiences confirmed that studying powders on the Franz diffusion cell was not possible, because the drug could not diffuse when it was placed on the surface of the membrane. Therefore, these results are not presented. During the investigation of nasal powders, the drug must dissolve before diffusion, which is feasible using the Side-Bi-Side cell, because of the horizontal orientation of the chambers [36]. Side-Bi-Side diffusion measurements (Figure 6) did not show a significant difference between MX/PVP mix powder and nanoMX/PVP powder. This can be explained by the aggregation of nanoparticles, which are controlled by surface forces and, if the particles are not stabilized, they may coagulate because of the high particle mobility. Presumably, further stabilization is needed for nanoparticles, when they are used in a powder form [37]. Considering the physical mixtures of MX and PVP, it can be concluded that the largest amount of drug was permeated from the powder form, due to its lowest viscosity compared to the other forms [38].

### 3.3. Evaluation of the Flux and the Permeability Coefficient

By both measurement methods, the flux (J), which shows the amount of MX that permeates through 1 cm^2^ of the membrane within 1 h, was significantly higher in the case of the nasal spray and gel forms, which contained nanoparticles compared to the formulations containing physical mixtures. An exceptionally high flux value was obtained for nanoMX gel using the Franz diffusion system. Concerning the liquid and semi-solid formulations, the permeability coefficients (K_p_)—calculated from the flux data—for the nanonized MX were also significantly higher than in the other cases (Table 3). The particle size was a determinant of the amount of diffused MX. The decrease of MX particle size to nano range resulted in nanoparticles with beneficial properties, like increased saturation solubility, dissolution rate, adhesivity to membranes, and consequently higher permeability of the drug [39].

## 4. Conclusions

The aim of this research was to compare the applicability of two diffusion models for permeability investigations of different intranasal formulations. A nanonized MX-containing co-ground product, and a physical mixture of MX and PVP were used to prepare spray, gel and powder forms. The drug was present in a suspended form in the liquid and semi-solid preparations. For each formulation, it can be noted that more MX diffused from the nanonized drug-containing samples, compared to those containing the physical mixture. This phenomenon could be explained by the amorphization, rapid dissolution and better distribution of nanoparticles.

The results support the theory of the suitability of the Side-Bi-Side horizontal cell to model intranasal administration in the case of some nasal dosage forms. During the comparative studies of spray, gel and powder forms, it was concluded that the Side-Bi-Side cell was adaptable to investigate drug diffusion from powders and liquids, and in particular, for the examination of suspensions. Due to the continuous stirring of the donor phase, a homogeneous distribution of the drug particles could be ensured. By monitoring permeation together with dissolution, better in vitro–in vivo correlation could be achieved [36]. It has to be noted, that nasal powders are becoming more and more important for nasal administration, because choosing this dosage form, the stability problems of some active ingredients could be eliminated. The measurement of the gel forms was not reliable in the horizontal system, because distribution was not uniform. In vitro measurements carried out by the Franz cell are current and well-known in the industry, nevertheless this method cannot imitate well the nasal conditions. It is proposed for the permeability measurements of nasal gels.

## Figures and Tables

**Figure 1 pharmaceutics-13-00846-f001:**
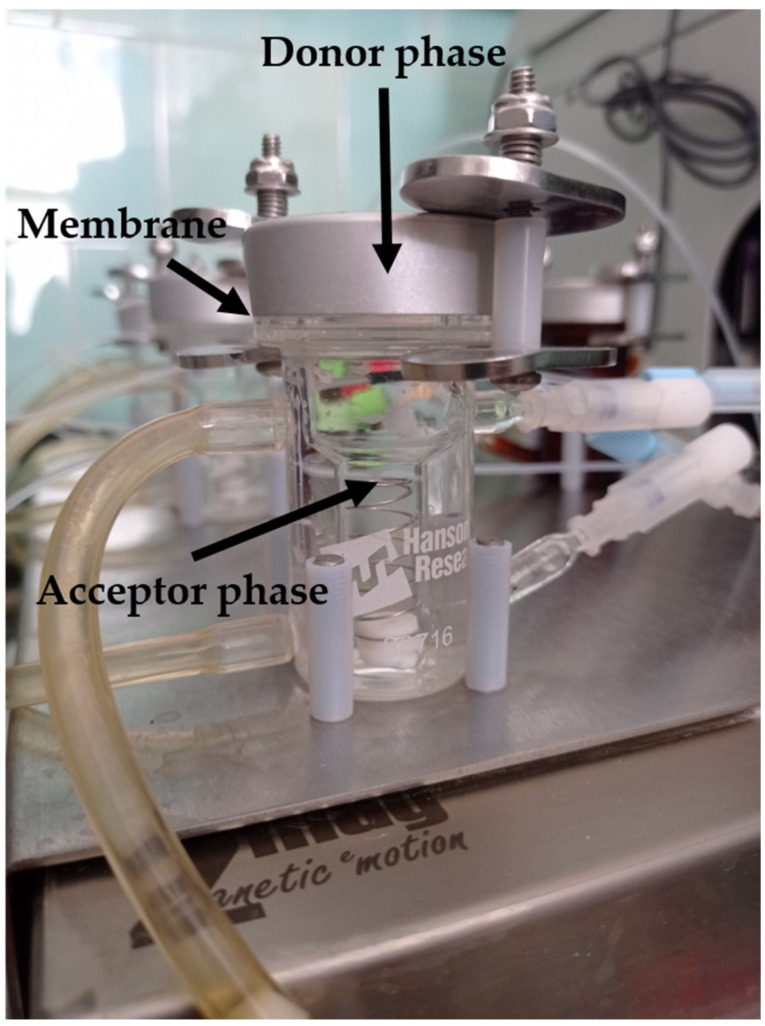
Illustration of Franz cell.

**Figure 2 pharmaceutics-13-00846-f002:**
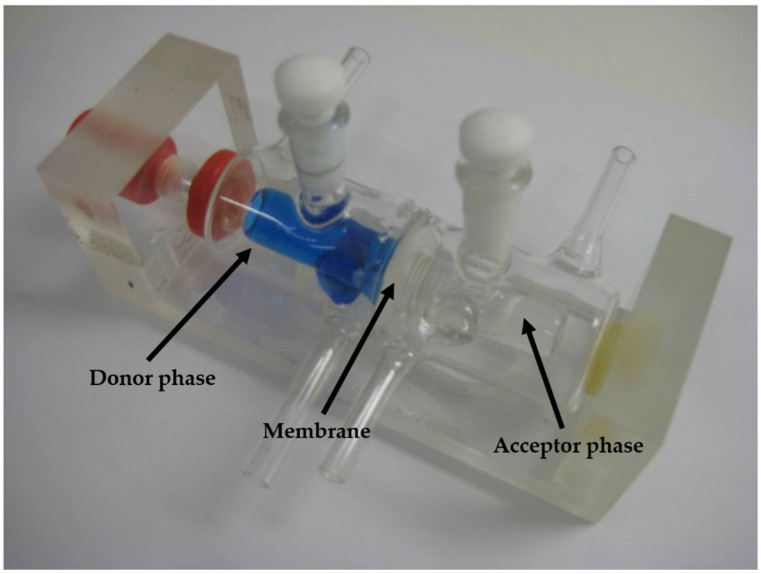
Illustration of Side-Bi-Side cell.

**Figure 3 pharmaceutics-13-00846-f003:**
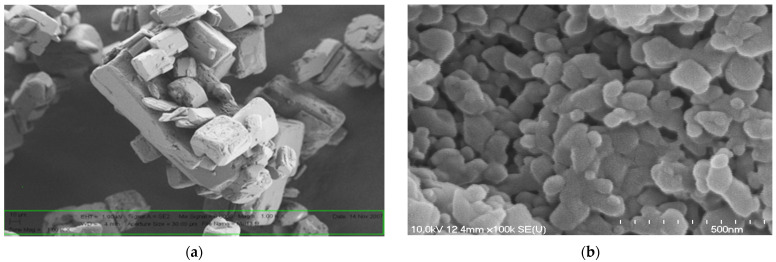
The SEM images of raw MX (**a**) and nanoMX powder (**b**).

**Figure 4 pharmaceutics-13-00846-f004:**
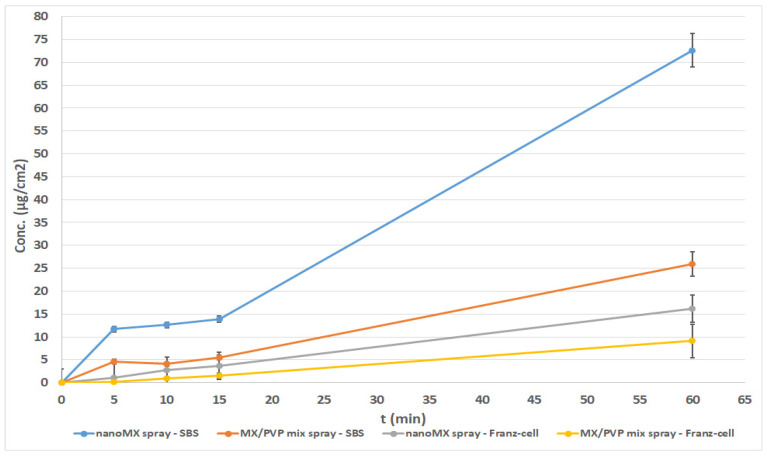
In vitro permeability of intranasal sprays on Franz and Side-Bi-Side diffusion systems.

**Figure 5 pharmaceutics-13-00846-f005:**
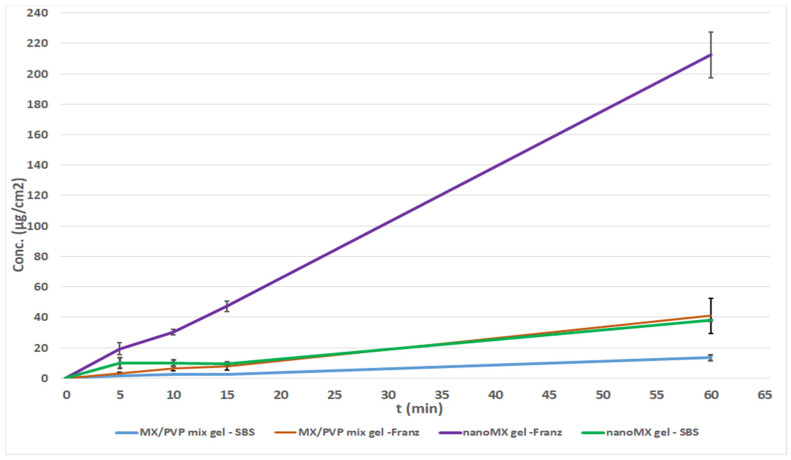
In vitro permeability of intranasal gels on Franz and Side-Bi-Side diffusion systems.

**Figure 6 pharmaceutics-13-00846-f006:**
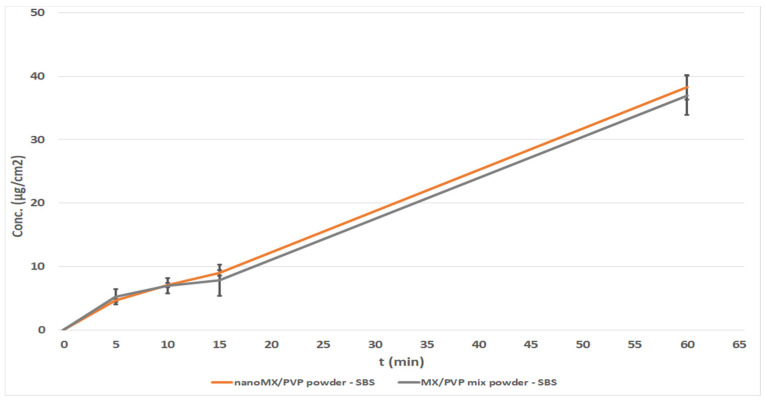
In vitro permeability of intranasal powders on Franz and Side-Bi-Side diffusion systems.

**Table 1 pharmaceutics-13-00846-t001:** Compositions of the intranasal formulations.

Sample	NanoMX (mg/mL)	Raw MX (mg/mL)	PVP (mg/mL)	HA (mg/mL)
NanoMX powder	1.0	-	1.0	-
NanoMX spray	1.0	-	1.0	1.0
NanoMX gel	1.0	-	1.0	5.0
MX/PVP mix powder	-	1.0	1.0	-
MX/PVP mix spray	-	1.0	1.0	1.0
MX/PVP mix gel	-	1.0	1.0	5.0

**Table 2 pharmaceutics-13-00846-t002:** Comparison of reliable differences of Side-Bi-Side cell and Franz cell.

Header	Franz Cell (Hanson Research Co., Los Angeles, CA, USA)	Side-Bi-Side Cell (Crown Glass, New York, NY, USA)
Relative position of the donor and acceptor chamber	vertical	horizontal
Surface of the diffusion	1.8 cm^2^	0.69 cm^2^
Volume of donor phase	7 mL	3 mL
Volume of acceptor phase	0.3 mL	3 mL
Place of sample administration	Directly to the membrane	Into the donor compartment

**Table 3 pharmaceutics-13-00846-t003:** Flux (J) and permeability coefficient (K_p_) values of intranasal formulations on Franz and Side-Bi-Side diffusion systems.

	Franz Cell		Side-Bi-Side-Cell	
Sample	J (µg/cm^2^/h)	K_p_ (cm/h)	J (µg/cm^2^/h)	K_p_ (cm/h)
NanoMX powder	-	-	38.26	0.03826
NanoMX spray	16.20	0.01620	72.61	0.07261
NanoMX gel	212.44	0.21244	37.97	0.03797
MX/PVP mix powder	-	-	36.96	0.03696
MX/PVP mix spray	9.14	0.00914	25.93	0.02593
MX/PVP mix gel	40.90	0.04090	13.48	0.01348

## Data Availability

The data presented in this study are available on request from the corresponding author.

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
