# Peer review of "Comparison of Modern In Vitro Permeability Methods with the Aim of Investigation Nasal Dosage Forms"

_pharmaceutics, 2021, doi:10.3390/pharmaceutics13060846_

Round 1

Reviewer 1 Report

Excellent paper aiming to study and compare a well- and a less-known in vitro permeability investigation method in order to ascertain which is suitable for determination of drug permeability through the nasal mucosa from different formulations. Well organized, well designed experiments and sound conclusions.

Author Response

Reply to Referee comments

For Reviewer 1

Thank you very much for your kind words.

  1. 05. 2021. Szeged, Hungary Csilla Balla-Bartos PhD

                                                                                                   Assistant Professor

Reviewer 2 Report

The authors describe two in vitro permeability methods in order to investigate nasal dosage forms. The paper is well described and references are quite updated. The experimental section is properly reported and described.

The authors report three kinds of formulations (spray forms, gel forms, and powder forms) observing a different behavior using the two analytic methods to evaluate the permeability. The authors also explained the reason of their results. However, the author use only one API (meloxicam, a poor water soluble FANS) and they conclude that the results depend only by the kind of formulation. Could  the nature of API affect the permeability measurements of nasal forms? On the contrary, do the author think that, independently of the API, they could obtain the same results changing the API in the formulations? A comparison with other drugs could improve the quality of the paper. 

The author should address this point. 

Author Response

Reply to Referee comments

For Reviewer 2

Thank you very much for the valuable remarks. We greatly appreciate your advices. Here you can see the modification made in the paper according to your suggestion (shown in blue colour in the text).

  1. The authors report three kinds of formulations (spray forms, gel forms, and powder forms) observing a different behavior using the two analytic methods to evaluate the permeability. The authors also explained the reason of their results. However, the author use only one API (meloxicam, a poor water soluble FANS) and they conclude that the results depend only by the kind of formulation. Could the nature of API affect the permeability measurements of nasal forms? On the contrary, do the author think that, independently of the API, they could obtain the same results changing the API in the formulations? A comparison with other drugs could improve the quality of the paper.

The author should address this point.

Thank you for your suggestion. The text was modified.

In view of the properties of nanonized meloxicam compared to the raw material (Kürti et al., 2013; Bartos et al., 2015; Horváth et al., 2016; Bartos et al., 2018), in this study the aim was to investigate and compare the permeability behavior of different dosage forms using two in vitro diffusion methods independently of the quality of drug.

The molecular weight, size, solubility, partition coefficient and pKa value affect the permeability measurements of nasal formulations. However, based on our other previous experiences, it can be noted that other active substances that are poorly (lamotrigine) or highly soluble (levodopa) in water showed a similar trend in diffusion studies comparing the drug with reduced particle size to the raw form (Ambrus et al., 2020; Bartos et al., 2018).

„Formulation factors like dosage form, pH, viscosity, mucoadhesivity also influence [12–14] the drug diffusion through the nasal mucosa. To overcome the short residence time of drugs in the nasal cavity resulted by mucociliary clearance, different mucoadhesive excipients (e.g. chitosans, lectins, thiomers, poloxamer or sodium hyaluronate) may be used during formulation, in order to prolong the contact time with the nasal mucosa, thereby enhancing the delivery of drugs [15–17]. Furthermore the usage of mucoadhesive additives allows the delivery of nanosized drugs intranasally overcoming the limitations of the particle size requirements for nasal formulations. Based on our previous experience it can be concluded that the bioavailability of an active pharmaceutical ingredient increased by decreasing its particle size [18,19].”

Kürti, L., Gáspár, R., Márki, Á., Kápolna, E., Bocsik, A., Veszelka, Sz., Bartos, Cs., Ambrus, R., Vastag, M., Deli, A. M., Szabó-Révész, P. In vitro and in vivo characterization of meloxicam nanoparticles designed for nasal administration. Eur. J. Pharm. Sci. 50 (2013) 86-92.

Bartos, C., Ambrus, R., Sipos, P., Budai-Szűcs, M., Csányi, E., Gáspár, R., Márki, Á., Seres, A. B., Sztojkov-Ivanov, A., Horváth, T., Szabó-Révész, P. Study of sodium hyaluronate-based intranasal formulations containing micro- or nanosized meloxicam particles. Int. J. Pharm. 491 (2015) 198-207.

Horváth, T., Ambrus, R., Völgyi, G., Budai-Szűcs, M., Márki, Á., Sipos, P., Bartos, C., Seres, A.B., Sztojkov-Ivanov, A., Takács-Novák, K., Csányi, E., Gáspár, R., Szabó-Révész, P. Effect of Solubility Enhancement on Nasal Absorption of Meloxicam. Eur. J. Pharm. Sci. 95 (2016) 96-102.

Bartos, C., Ambrus, R., Kovács, A., Gáspár, R., Sztojkov-Ivanov, A., Márki, Á., Janáky, T., Tömösi, F., Kecskeméti, G., Szabó-Révész, P. Investigation of Absorption Routes of Meloxicam and Its Salt Form from Intranasal Delivery Systems. Molecules 23 (2018) 784.

[18] „Ambrus, R., Gieszinger, P., Gáspár, R., Sztojkov-Ivanov, A., Ducza, E., Márki, Á., Janáky, T., Tömösi, F., Kecskeméti, G., Szabó-Révész, P., Bartos, Cs. Investigation of the Absorbtion of Nanosized Lamotrigine Containing Nasal Powder via the Nasal Cavity. 25 (2020) 1065.

[19] Bartos, Cs., Pallagi, E., Szabó-Révész, P., Ambrus, R., Katona, G., Kiss, T., Rahimi, M., Csóka, I. Formulation of levodopa containing dry powder for nasal delivery applying the quality-by-design approach. Eur. J. Pharm. Sci. 123 (2018) 475-483.”

  1. 05. 2021. Szeged, Hungary Csilla Balla-Bartos PhD

                                                                                                   Assistant Professor

Reviewer 3 Report

The authors compared the suitability for determination of drug permeability across the nasal mucosa between the vertical diffusion cell (Franz-cell) and the horizontal diffusion model (Side-Bi-Side™). I agree with the authors that it is essential to establish an in vitro diffusion and dissolution tests especially for investigation nasal of formulations. However, I don’t find the significance of the study. The conclusions are apparent. Besides, the study didn’t facilitate the establishment of the in vitro dissolution test for nasal preparations.

  1. It is of minor significance to evaluate the suitability of Side-Bi-Side™ cell for in vitro evaluation of nasal preparations, because the instrument fails to mimic the nasal environment. Actually, the horizontal diffusion model is suitable to evaluate the permeability of drug molecules instead of the preparations.
  2. It is obvious that the horizontal diffusion model is not suitable for the evaluation of powders.
  3. In addition to the comparison between different models, it is more important to find an alternative membrane for nasal mucusa.
  4. What are the significance for comparison between nano-preparations and the mixed ones? Are the results facilitating the establishment of the in vitro dissolution methods? Instead, it is critical to compare the in vitro results with the in vivo ones to validate the feasibility of the model.

Author Response

Reply to Referee comments

For Reviewer 3

Thank you very much for the valuable remarks. We greatly appreciate your advices. Here you can see listed the modifications made in the paper according to your suggestions (shown in red colour in the text).

List of corrections:

  1. It is of minor significance to evaluate the suitability of Side-Bi-Side™ cell for in vitro evaluation of nasal preparations, because the instrument fails to mimic the nasal environment. Actually, the horizontal diffusion model is suitable to evaluate the permeability of drug molecules instead of the preparations.

Thank you for your suggestion. The introduction was modified.

„A horizontal Side-Bi-Side system has a small volume of donor and acceptor compartments allowing the measurement of small amounts of samples, as nasal dosage forms. Magnetic stirring of the donor phase imitates the movement of cilia in the nasal cavity [26].”

The Side-Bi-Side™ can be used for blood-brain barrier and nasal experiments (Horváth et al., 2015). The volume of the acceptor and donor compartments are small (3ml), therefore it is proper for the investigation of small amounts of samples. Magnetic stirring is possible in the donor phase imitating the movement of cilia. In case of preparations the dissolution and diffusion of drug are investigated together in one step.

Horváth, T., Ambrus, R., Szabó-Révész, P. Investigation of permeability of intranasal formulations using Side-Bi-Side horizontal diffusion cell. Acta Pharm. Hung. 85 (2015) 19-28.

  1. It is obvious that the horizontal diffusion model is not suitable for the evaluation of powders.

Thank you for your comment.

Side-Bi-Side™ cell was adaptable to investigate the drug diffusion from powders and liquids, and especially for the examination of suspensions. Thanks to the continuous stirring of donor phase, a homogeneous distribution of the drug particles could be ensured. Investigation of powders on Franz diffusion cell was not possible, because the drug could not diffuse when it was placed on the surface of the membrane.

  1. In addition to the comparison between different models, it is more important to find an alternative membrane for nasal mucusa.

Thank you for your suggestion. The text was modified.

„For the investigaiton of nasal formulations screening methods are used which are able to give information about nasal preparations. A general penetration investigation protocol of a pharmaceutical nasal composition includes in vitro, in vitro cell line, ex vivo and in vivo investigations. In vitro studies are carried out using artificial membrane and the development of in vitro models is of great importance in simplification and quickening of animal studies [20]. Several solutions and methods were carried out to imitate the nasal environment and help to predict the behaviour of the tested forms…”

Development of in vitro models is very important from economic and ecological aspects because with their help, pharmaceutical development can be radically accelerated by reducing the time and other sources invested in animal studies (Fallacara et al., 2019). The aim of our study was to compare two different cell systems using the same artificial membrane from the aspect of their applicability for investigation of nasal formulations.

[20] Fallacara, A., Busato, L., Pozzoli, M., Ghadiri, M., Ong, H.X., Young, P.M., Manfredini, S., Traini, D. In vitro characterization of physico-chemical properties, cytotoxicity, bioactivity of urea-crosslinked hyaluronic acid and sodium ascorbyl phosphate nasal powder formulation. Int. J. Pharm. 558 (2019) 341-350.

  1. What are the significance for comparison between nano-preparations and the mixed ones? Are the results facilitating the establishment of the in vitro dissolution methods? Instead, it is critical to compare the in vitro results with the in vivo ones to validate the feasibility of the model.

Thank you for your suggestion.

The physical mixture containing samples were applied as the control samples, however they can also provide information on the applicability of diffusion methods for certain forms. Whereas our previous experiences have shown that the reduction of drug particle size resulted in better permeability comparing with the raw form (Ambrus et al., 2020; Bartos et al., 2018). Therefore if this trend changes, there can be a problem with the permeability investigation (e.g. uneven distribution or aggregation of particles). As a first step of permeation protocol we investigated the in vitro methods. Our further aims include the study of the entire investigation protocol, and as its last point the in vivo investigation as well.

„Ambrus, R., Gieszinger, P., Gáspár, R., Sztojkov-Ivanov, A., Ducza, E., Márki, Á., Janáky, T., Tömösi, F., Kecskeméti, G., Szabó-Révész, P., Bartos, Cs. Investigation of the Absorbtion of Nanosized Lamotrigine Containing Nasal Powder via the Nasal Cavity. 25 (2020) 1065.

Bartos, Cs., Pallagi, E., Szabó-Révész, P., Ambrus, R., Katona, G., Kiss, T., Rahimi, M., Csóka, I. Formulation of levodopa containing dry powder for nasal delivery applying the quality-by-design approach. Eur. J. Pharm. Sci. 123 (2018) 475-483.”

  1. 05. 2021. Szeged, Hungary Csilla Balla-Bartos PhD

                                                                                                   Assistant Professor

Reviewer 4 Report

The current manuscript entitles “Comparison of modern in vitro permeability methods with the aim of investigation nasal dosage forms” by Bartos et al., is an admirable attempt to compare vertical and horizontal diffusion cells to study the permeation performance of nasal formulations. Although the overall manuscript is nicely presented, I shall suggest that authors should consider the following comments and consequently modify the manuscript.

Major comments:
-         The overall rationale section is weak- please consider clarifying your message. As you have described various systems available to study drug permeation studies, why vertical and horizontal diffusion cells were chosen for comparison? Please update this in the abstract too.
Please add a sentence “take-home message” at the end of ABSTRACT, so the reader should be aware of what you are concluding remarks with possible advantages.
-           Section 3.2.1, 3.2.2 and 3.2.3 all are descriptive (dominantly explaining the results). Please consider discussing the results backed by scientific explanation/theories and appropriate literature to cater for the “WHY” question?
-         A proof-reading is strongly advised
Minor comments:
-         At various places, authors have used the word “thanks to” I am not sure, but I think it is not a very good idea in scientific, academic writing, so please consider revising with some scientific explanation instead “thanks to”

Author Response

Reply to Referee comments

For Reviewer 4

Thank you very much for the valuable remarks. We greatly appreciate your advices. Here you can see listed the modifications made in the paper according to your suggestions (shown in green colour in the text and with red colour for Reviewer 3). English language of manuscript was checked.

  1. The overall rationale section is weak- please consider clarifying your message. As you have described various systems available to study drug permeation studies, why vertical and horizontal diffusion cells were chosen for comparison? Please update this in the abstract too.
    Please add a sentence “take-home message” at the end of ABSTRACT, so the reader should be aware of what you are concluding remarks with possible advantages.

Thank you for your suggestion, the text is modified.

„Franz diffusion cell system is the official Pharmacopoeial method primarily for investigation of diffusion of transdermal formulations, but it is one of the most commonly used method for investigation of intransal dosage forms in the literature. [24,25]. A horizontal Side-Bi-Side system has a small volume of donor and acceptor compartments allowing the measurement of small amounts of samples, as nasal dosage forms. Magnetic stirring of the donor phase imitates the movement of cilia in the nasal cavity [26].

Our aim was to compare two in vitro investigation models with different orientation of the phases; as a vertical system the Franz diffusion cell and as a horizontal one the Side-Bi-SideTM system from the aspect of their applicability for investigation of different intranasal dosage forms: sprays, gels and powders. Nasal formulations contained MX as an active pharmaceutical ingredient (API).”

The abstract is modified also.

„It can be concluded that the application of a horizontal cell is recommended for liquid and solid nasal preparations, while the vertical one should be used for semi-solid formulations.”

  1. Section 3.2.1, 3.2.2 and 3.2.3 all are descriptive (dominantly explaining the results). Please consider discussing the results backed by scientific explanation/theories and appropriate literature to cater for the “WHY” question?

3.2.1. Investigation of nasal spray forms

The results of the permeability measurements of the spray forms with the same content (nanoMX spray and MX/PVP mix spray) were compared on Franz diffusion cell system and Side-Bi-Side system (Figure 4). It can be seen, that the highest diffused amount of MX (≈73 µg/cm2) and the quickest diffusion (due to the rapid dissolution of the drug) were observed in case of nanoMX spray, measured on the horizontal diffusion cell. Initially, a rapid rising in the concentration was noticed (in first 5 min), followed by a less steeply increasing drug amount in the acceptor phase. The diffusion of nanoMX spray was at least approximately two times higher than the others in 60 minutes. Beyond the better dissolution of amorphized nanoparticles [29,30], this observation can be explained by the uniform distribution of MX particles in the continuously stirred liquid medium, containing 1 mg/ml HA, as a mucoadhesive agent. The pH of the formulaiton (pH 5.6) and the low concentration of HA resulted in a sol state of samples. It should be excluded that the nanoMX particles (D0.5=140nm) passed through the membrane without dissolving, because the membrane pore size (100nm) was smaller than the particle size and the membrane was impregnated with isopropyl myristate [7]. The measurement of spray forms was difficult on Franz diffusion cell. Because of the low viscosity of the samples, keeping them on the horizontal surface of membrane was problematic. Since, stirring is not resolved – as mentioned above – the measurement of suspensions and sedimenting materials was limited. The results showed steady increase in concentration and lower permeated amount of MX comparing with results measured on Side-Bi-Side system. This phenomenon could be explain partly with the fact that Franz cell was developed for the investigation of semi-solid forms. On the other hand, the donor phase of Side-Bi-SideTM cell is suitable for inserting liquid formulations and due to the stirring with a magnetic stirrer, it can be useful for testing liquids and suspensions.

[29] Hu, J., Johnston, K.P., Williams, R. O. Nanoparticle Engineering Processes for Enhancing the Dissolution Rates of Poorly Water Soluble Drugs. Drug Develop. Ind. Pharm. 30 (2004) 233-245.

[30] Alonzo, D. E., Zhang, G. G. Z., Zhou, D., Gao, Y., Taylor, L. S. Understanding the Behavior of Amorphous Pharmaceutical Systems during Dissolution. Pharm. Res. 27 (2010) 608-618.

[7] Bartos, C., Ambrus, R., Sipos, P., Budai-Szűcs, M., Csányi, E., Gáspár, R., Márki, Á., Seres, A.B., Sztojkov-Ivanov, A., Horváth, T. et al. Study of Sodium Hyaluronate-Based Intranasal Formulations Containing Micro- or Nanosized Meloxicam Particles. Int. J. Pharm. 491 (2015) 198-207.

3.2.2. Investigation of gel forms

Permeability of gels with the same content (nanoMX gel and MX/PVP mix gel) were also compared on Franz diffusion cell system and Side-Bi-Side system. Studying the samples, it can be seen that much higher permeated drug concentration was determined in case of nanoMX gel applying Franz diffusion system (215 µg/cm2) (Figure 5). Almost ten-fold drug amount was diffused to the acceptor phase comparing to the results of the measurements on the Side-Bi-Side cell, which could be explained, on the one hand by the rapid dissolution of nanosized and amorphized MX particles [31], which provided a faster diffusion and a higher drug concentration in the acceptor phase. On the other hand, the design of Franz-cell’s donor phase was developed for the investigation of semi-solid formulations [32,33]. The permeated drug amount of nanoMX gel on Side-Bi-Side system was similar to the results of the physical mixture investigated on Franz-cell, and the lowest drug amount was detected in case of MX/PVP mix gel on Side-Bi-Side diffusion cell. This ob-servation can be explained by the inadequate stirring in the donor compartment, furthermore nanoMX has a particle size similar to those of polymeric molecules such as HA, PVP which can result in a well-structured complex, and better interactions among the components retaining MX from dissolution and prevented the drug to get to the membrane surface [34,35]. It can be concluded that Franz diffusion system is proposed for investigation of intranasal gels.

[31] Dizaj, S. M., Vazifehas, Zh., Salatin, S., Adibkia, Kh., Javadzadeh, Y. Nanosizing of drugs: Effect on dissolution rate. Res. Pharm. Sci. 10 (2015) 95-108.

[32] Salamanca, C. H., Barrera-Ocampo, A., Lasso, J. C., Camacho, N., Yarce, C. J. Franz Diffusion Cell Approach for Pre-Formulation Characterisation of Ketoprofen Semi-Solid Dosage Forms. Pharmaceutics 10 (2018) 148.

[33] Ng, S-F., Rouse, J. J., Sanderson, F. D., Meidan, V., Eccleston, G. M. Validation of a Static Franz Diffusion Cell System for In Vitro Permeation Studies. AAPS Pharm. Sci. Tech. 11 (2010) 1432-1441.

[34] Bartos, Cs. Application of wet milling techniques to produced micronized and nanonized drug pre-dispersions for the development of intranasal formulations. PhD thesis 2016. University of Szeged

[35] Battistini, F. D. Eugenia, M., Rubén. O., Manzo. H. Equilibrium and release properties of hyaluronic acid–drug complexes. Eur. J. Pharm. Sci. 49 (2013) 588-594.

3.2.3. Investigation of powder forms

Our experiences confirmed that studying powders on Franz diffusion cell was not possible, because the drug could not diffuse when it was placed on the surface of the membrane. Therefore, these results are not presented. During the investigation of nasal powders, the drug must dissolve before diffusion which is feasible using Side-Bi-Side cell, because of the horizontal orientation of chambers [36] . Side-Bi-Side diffusion measurements (Figure 6) did not show significant difference between MX/PVP mix powder and nanoMX/PVP powder. This can be explained by the aggregation of nanoparticles, which are controlled by surface forces and,if the particles are not stabilized, they may coagulate because of the high particle mobility. Presumably, further stabilization is needed for nanopartcles, when they are used in a powder form [37]. Considering the physical mixtures of MX and PVP, it can be concluded that the largest amount of drug was permeated from the powder form, due to the lowest viscosity compared to the other forms [38].

[36] Borbás, E., Balogh, A. Bocz, K. Müller, J., Kiserdei, É., Vigh, T., Sinkó, B., Marosi, A., Halász, A., Dohányos, Z. et al. In Vitro Dissolution–Permeation Evaluation of an Electrospun Cyclodextrin-Based Formulation of Aripiprazole Using ΜFluxTM. Int. J. Pharm. 491 (2015) 180-189.

[37] Paltonen, L., Hirvonen, J. Pharmaceutical nanocrystals by nanomilling: critical process parameters, particle fracturing and stabilization methods. J. Pharm. Pharmacol. 62 (2010) 1569-1579.

[38] Obeidat, W. M., Price, J. C. Viscosity of polymer solution phase and other factors controlling the dissolution of theophylline microspheres prepared by the emulsion solvent evaporation method. J. Microencapsulation. 20 (2003) 57-65.

  1.   A proof-reading is strongly advised.

Thank you for your suggestion, proof-reading was carried out. The manuscrpit was corrected.

  1. At various places, authors have used the word “thanks to” I am not sure, but I think it is not a very good idea in scientific, academic writing, so please consider revising with some scientific explanation instead “thanks to”

Thank you for your suggestion, the words „thanks to” were modified to „due to” throughout the manuscript.

  1. 05. 2021. Szeged, Hungary Csilla Balla-Bartos PhD

                                                                                                   Assistant Professor

Round 2

Reviewer 4 Report

Suggestions have been satisfactorily adopted.